

# Characterization of the complete chloroplast genomes of five *Populus* species from the western Sichuan plateau, southwest China: comparative and phylogenetic analyses

Dan Zong[1,2], Anpei Zhou[1,2], Yao Zhang[1,2], Xinlian Zou[1,2], Dan Li[3], Anan Duan[1,2,4] and Chengzhong He[1,2,4]

[1] Key Laboratory for Forest Genetic and Tree Improvement & Propagation in Universities of Yunnan Province, Southwest Forestry University, Kunming, Yunnan, China
[2] Key Laboratory of State Forestry Administration on Biodiversity Conservation in Southwest China, Southwest Forestry University, Kunming, Yunnan, China
[3] Yunnan Academy of Biodiversity, Southwest Forestry University, Kunming, Yunnan, China
[4] Key Laboratory for Forest Resources Conservation and Use in the Southwest Mountains of China, Ministry of Education, Southwest Forestry University, Kunming, Yunnan, China

Corresponding author
Chengzhong He, hcz70@163.com

## ABSTRACT

Species of the genus *Populus*, which is widely distributed in the northern hemisphere from subtropical to boreal forests, are among the most commercially exploited groups of forest trees. In this study, the complete chloroplast genomes of five *Populus* species (*Populus cathayana*, *P. kangdingensis*, *P. pseudoglauca*, *P. schneideri,* and *P. xiangchengensis*) were compared. The chloroplast genomes of the five *Populus* species are very similar. The total chloroplast genome sequence lengths for the five plastomes were 156,789, 156,523, 156,512, 156,513, and 156,465 bp, respectively. A total of 130 genes were identified in each genome, including 85 protein-coding genes, 37 tRNA genes and eight rRNA genes. Seven genes were duplicated in the protein-coding genes, whereas 11 genes were duplicated in the RNA genes. The GC content was 36.7% for all plastomes. We analyzed nucleotide substitutions, small inversions, simple sequence repeats and long repeats in the chloroplast genomes and found nine divergence hotspots (*ccsA+ccsA-ndhD, ndhC-trnV, psbZ-trnfM, trnG-atpA, trnL-ndhJ, trnR-trnN, ycf4-cemA, ycf1,* and *trnR-trnN*), which could be useful molecular genetic markers for future population genetic and phylogenetic studies. We also observed that two genes (*rpoC2* and *rbcL*) were subject to positive selection. Phylogenetic analysis based on whole cp genomes showed that *P. schneideri* had a close relationship with *P. kangdingensis* and *P. pseudoglauca*, while *P. xiangchengensis* was a sister to *P. cathayana*.

## INTRODUCTION

The species of the genus *Populus*, collectively known as poplar, are widely distributed in the northern hemisphere from subtropical to boreal forests and one of the most

commercially exploited groups of forest trees (*Hamzeh & Dayanandan, 2004*). Because of their small genome size, fast growth rates, profuse vegetative propagation, adaptability to a variety of ecological sites, and their wood's numerous uses, *Populus* species have become one of the most economically important groups of forest trees and a model organism for the study of tree biology (*Braatne, Hinckly & Stettler, 1992*; *Stettler, Zsuffa & Wu, 1996*). According to a recent classification, the genus *Populus* is classified into six sections (*Fang, Zhao & Skvortsov, 1999*; *Zsuffa, 1975*; *Eckenwalder, 1996*). To date, more than 100 *Populus* species or varieties have been reported worldwide, of which approximately 53 are endemic to China (*Wan & Zhang, 2013*).

As a concentrated area of *Populu*s resources in southwest China, the western Sichuan Plateau is dominated by mountainous and plateau geomorphology, and the mountains play a critical role in isolating plant distribution (*He et al., 2015*). Meanwhile, the complex and unique natural and geographical conditions of this area provide not only diversified refuges where plants retreat in response to climatic changes but also great opportunities to develop new hybrid species (*Lu et al., 2014*). However, the extensive interspecific hybridization and the high levels of morphological variation in *Populus* have posed great difficulties in species delimitation for systematic and comparative evolutionary studies (*Hamzeh & Dayanandan, 2004*; *Eckenwalder, 1996*; *Cronk, 2005*).

*Populus kangdingensis*, *P. pseudoglauca*, *P. schneideri*, and *P. xiangchengensis* are native to the western Sichuan Plateau, and they are distributed at altitudes above 3,000 m and even above 4,000 m, whereas *P. cathayana* widely occurs in China at altitudes ranging from 800 to 3,000 m. All five species overlap in the western Sichuan Plateau. Previous research has focused on their phylogenetic relationships. *Liu & Fu (2004)* considered *P. xiangchengensis* a hybridization of *P. schneideri* and *P. pseudoglauca* based on morphological characteristics, while another study suggested that *P. xiangchengensis* was a likely hybrid species of *P. kangdingensis* and *P. pseudoglauca* based on morphological characteristics (*Wan et al., 2009*). *P. schneideri* was classified into section *Tacamahaca*. Meanwhile, it was also considered a natural hybrid formed by *P. kangdingensis* and *P. cathayana* based on inter-simple sequence repeat (ISSR) and internal transcribed spacer (ITS) molecular markers (*Chen et al., 2007*; *Wang, 2012*). *P. pseudoglauca* was originally classified in section *Leucoides*, although it was suggested to be assigned to section *Tacamahaca* (*Zhao, 1994*), and this assignment was supported by ISSR and nuclear ITS sequence markers (*Wang, 2012*). The relationship between the five *Populus* species is shown in Fig. 1. All these findings suggested that the phylogenetic relationship of the five *Populus* species is rather complex and unclear.

Organellar DNA, which has its own genome and is inherited from one parent, is well conserved and allows for the development of informative universal markers (*Howe et al., 2003*; *Wicke et al., 2011*). The chloroplast (cp) genome, because of its relatively conserved size, gene content, structure and slow rate of nucleotide substitution within protein-coding genes, has been an ideal source of data on the phylogenetic relationships of plant taxa and their evolution and has been used to make significant contributions concerning evolutionary mechanisms for species and phylogenetic reconstruction (*Khan et al., 2012*; *Asheesh & Vinay, 2012*; *Liu et al., 2017*).

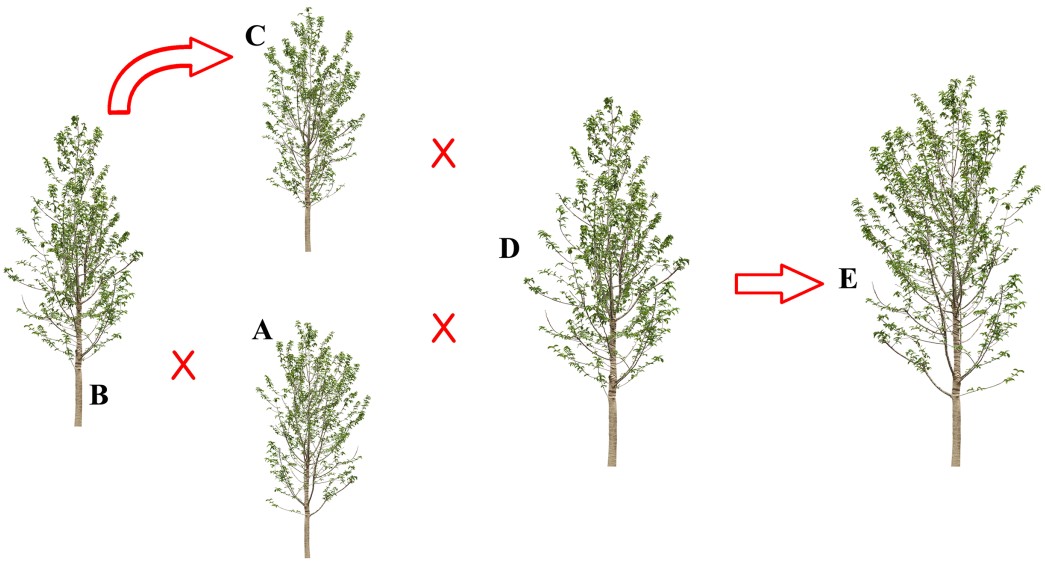

**Figure 1 The hybridization schematic diagram of five *Populus* species.** (A) *P. kangdingensis*; (B) *P. cathayana*; (C) *P. schneideri*; (D) *P. pseudoglauca*; (E) *P. xiangchengensis*. *P. schneideri* = *P. kangdingensis* × *P. cathayana*, *P. xiangchengensis* = *P. kangdingensis* × *P. pseudoglauca*, *P. xiangchengensis* = *P. schneideri* × *P. pseudoglauca*.

With the development of sequencing technology in recent years, in addition to nuclear genome sequences, cp genes, gene spacer regions, and cp genome sequences have been widely used to study plant molecular systematics. Whole cp genomes of several species from the genus *Populus* have been sequenced and deposited in GenBank. Here, we compare the complete cp genomes of *P. cathayana* (GenBank accession number: MK267318), *P. kangdingensis* (GenBank accession number: MK267310), *P. pseudoglauca* (GenBank accession number: MK267308), *P. schneideri* (GenBank accession number: MK267305), all sequenced in our study (*Zong et al., 2019a*), and *P. xiangchengensis* (GenBank accession number: MH910611, also found in the Supplemental Information) (*Zong et al., 2019b*). The codon usage bias, sequence divergences, mutation events, single nucleotide polymorphism (SNP) patterns and simple sequence repeat (SSR) distributions are compared, and a phylogenetic tree is reconstructed based on 27 complete cp genome sequences from Salicaceae. Our study provides cp genomic information for further phylogenetic reconstruction, molecular evolution research, and resources for breeding the genus *Populus*.

# MATERIALS AND METHODS

## Plant materials and DNA extraction

The fresh leaves of *P. cathayana* were collected in Kangding (101°56′26″E, 29°59′36″N, Sichuan, China; altitude: 3,109 m), while the samples of *P. kangdingensis*, *P. pseudoglauca*, *P. schneideri* and *P. xiangchengensis* were collected in Kangding (101°36′43″E, 30°05′20″N, Sichuan, China; altitude: 3,554 m), Yajiang (100°54′06″E, 29°59′14″N, Sichuan, China; altitude: 3,598 m), Litang (101°36′43″E, 30°05′20″N, Sichuan, China;

altitude: 4,018 m) and Xiangcheng (99º40′33″E, 28º55′47″N, Sichuan, China; altitude: 3,530 m), respectively. The voucher specimens of the five species were deposited at the herbarium of Southwest Forestry University, Kunming, China. Total genomic DNA was extracted with the Ezup plant genomic DNA prep kit (Sangon Biotech, Shanghai, China), and DNA samples were stored at −80 °C at the Key Laboratory of State Forestry Administration on Biodiversity Conservation in Southwest China, Southwest Forestry University, Kunming, China.

## Genome sequencing, assembly and annotation

Total DNA was used to generate libraries with an average insert size of 400 bp, which were sequenced using the Illumina HiSeq X platform. Approximately 15.0 GB of raw data were generated from each genome with 150 bp paired-end read lengths. Then, the raw data were used to assemble the complete cp genome using GetOrganelle software (*Jin et al., 2018*) with *P. trichocarpa* as the reference. Genome annotation was performed with the program Geneious R8 (Biomatters Ltd, Auckland, New Zealand) by comparing the sequences with the cp genome of *P. trichocarpa*. The tRNA genes were further confirmed through online tRNAscan-SE web servers (*Schattner, Brooks & Lowe, 2005*). A gene map of the annotated *Populus* cp genome was drawn by OGdraw online (*Lohse et al., 2013*).

## Indices of codon usage

As an important indicator of codon usage bias, the relative synonymous codon usage (RSCU) value is the frequency observed for a codon divided by its expected frequency (*Sharp, Tuohy & Mosurski, 1986*; *Sharp & Li, 1987*). The amino acid compositions and RSCU values of the five *Populus* cp genomes were calculated using the CodonW program (*Peden, 1999*). Because short CDSs generally result in large estimation errors for codon usage, CDSs shorter than 300 bp in length were excluded in codon usage calculations to avoid sampling bias (*Rosenberg, Subramanian & Kumar, 2003*). Finally, 58 CDSs from the five cp genomes were analyzed in this study.

## Genome comparison

To investigate divergence in cp genomes, identity across the whole cp genomes was visualized using the mVISTA viewer in Shuffle-LAGAN mode for the five species, with the *P. xiangchengensis* genome as the reference. MAFFT version 7 software (*Katoh et al., 2005*) was used to align the five plastome sequences, followed by adjustment with BioEdit. To elucidate the level of sequence variation, we then performed sliding window analysis to assess the pairwise variability (*Pi*) over the plastomes in DnaSP version 5 software (*Librado & Rozas, 2009*). The window length was set to 600 bp, and the step size was set to 200 bp. The SNP variation was detected using the "find variation" function in Geneious R8.

## Identification of simple sequence repeats and long sequence repeats

Simple sequence repeats in five *Populus* cp genomes were detected using Microsatellite identification tool (MISA) (*Thiel, Michalek & Varshney, 2003*) with the minimal repeat number set to 12, 6, 5, 5, 5, and 5 for mono-, di-, tri, tetra-, penta-, and hexa nucleotide sequences, respectively. We used the online REPuter software to identify and locate

forward (F), reverse (R), complemented (C), and palindromic (P) repeats. The following settings for repeat identification were used: (1) Hamming distance equal to 3; (2) minimal repeat size was set to 30 bp; (3) maximum computed repeats was set to 90 bp (*Kurtz et al., 2001*).

### Gene selective pressure analysis of five *Populus* plastomes

To examine variation in the evolutionary rates of cp genes, we calculated the nonsynonymous substitution rates ($Ka$), synonymous rates ($Ks$), and their ratios ($Ka/Ks$) using model averaging in the $Ka\_Ks$ Calculator program according to the LWL85 method (*Yang & Bielawski, 2000*; *Zhang et al., 2006*).

### Phylogenetic analysis

To explore the genetic relationships among the five species of the *Populus* genus, a total of 17 complete cp genomes of *Populus* and five plastomes of *Salix* were obtained from GenBank, and *Itoa orientalis* and *Idesia polycarpa* were used as the outgroups (Table S1). To examine the phylogenetic utility of different regions, phylogenetic analyses were performed based on the following data: (1) the complete cp DNA sequences, (2) the large single copy (LSC) region, (3) the small single copy region (SSC), (4) one inverted repeat region (IR), (5) the LSC+SSC region, (6) the LSC+SSC+IR region, and (7) a set of 85 common protein coding genes. All of the datasets were aligned using MAFFT under default settings. jModelTest 2.0 (*Darriba et al., 2012*) was used to determine the best-fitting model for each dataset based on the Akaike information criterion. A maximum likelihood method for phylogenetic analysis was performed based on the GTR+I+G model in RAxML version 8 (*Stamatakis, 2014*).

## RESULTS

### Features of the five *Populus* plastomes

The complete cp genomes of the five *Populus* species ranged from 156,465 (*P. xiangchengensis*) to 156,789 bp (*P. cathayana*) in length. The plastome size of *P. schneideri* was only one bp larger than that of *P. pseudoglauca*. The plastome size of *P. kangdingensis* was 11 bp larger than that of *P. pseudoglauca* and 166 bp smaller than that of *P. cathayana* (Table 1; Fig. 2). The five cp genomes included a pair of IRs of 27,620 bp in the three species *P. kangdingensis*, *P. pseudoglauca* and *P. schneideri* and an IR pair of 27,672 bp in *P. cathayana* and 27,570 bp in *P. xiangchengensis*. The GC contents were consistent in *P. kangdingensis*, *P. pseudoglauca*, and *P. schneideri*, with 34.5%, 30.5% and 42.0% in the LSC, small short SSC and IR regions, respectively (Tables 1 and 2).

Each of the *P. cathayana*, *P. kangdingensis*, *P. pseudoglauca*, *P. schneideri*, and *P. xiangchengensis* cp genomes encoded 130 functional genes; 112 of these were unique genes, including 78 protein-coding genes, 30 tRNA genes and four rRNA genes (Table S2). Most of these genes occurred as a single copy, while 18 genes were double copies: seven protein-coding genes, seven tRNA genes and four rRNA genes. The LSC region contained 59 protein-coding genes and 22 tRNA genes, whereas the SSC region contained 10 protein-coding genes and one tRNA gene.

**Table 1  The features of five *Populus* plastomes.**

| Species | Size (bp) | LSC (bp) | SSC (bp) | IR (bp) | Number of proteincoding genes | Number of tRNA genes | Number of rRNA genes | GC content (%) |
|---|---|---|---|---|---|---|---|---|
| *P. cathayana* | 156,789 | 84,851 | 16,594 | 27,672 | 85(7) | 37(7) | 8(4) | 36.7 |
| *P. kangdingensis* | 156,523 | 84,788 | 16,495 | 27,620 | 85(7) | 37(7) | 8(4) | 36.7 |
| *P. pseudoglauca* | 156,512 | 84,777 | 16,495 | 27,620 | 85(7) | 37(7) | 8(4) | 36.7 |
| *P. schneideri* | 156,513 | 84,778 | 16,495 | 27,620 | 85(7) | 37(7) | 8(4) | 36.7 |
| *P. xiangchengensis* | 156,465 | 84,812 | 16,513 | 27,570 | 85(7) | 37(7) | 8(4) | 36.7 |

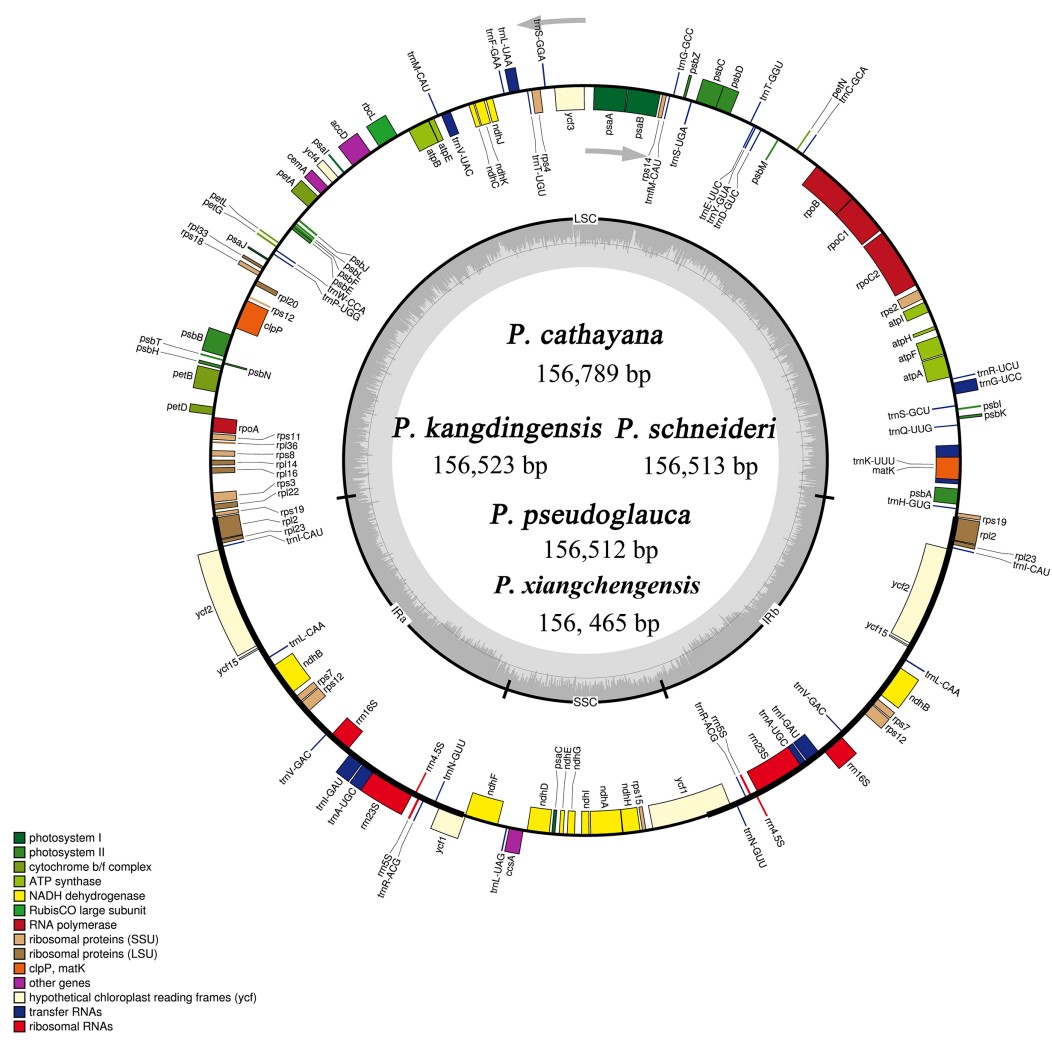

**Figure 2  Gene map of the five *Populus* species cp genomes.** The genes that are drawn outside of the circle are transcribed clockwise, whereas those that are drawn inside the circle are transcribed counterclockwise. Large single copy (LSC), small single copy (SSC) and inverted repeats (IRa and IRb) are indicated.                                                

## Codon usage

Most protein-coding genes had the standard AUG sequence as the start codon, but *ndhD* started with GUG, and *rpl16* started with ATC. ATC as an initiation codon has been

**Table 2 Base composition of the five _Populus_ plastomes.**

| Region | | _P. cathayana_ | _P. kangdingensis_ | _P. pseudoglauca_ | _P. schneideri_ | _P. xiangchengensis_ |
|---|---|---|---|---|---|---|
| LSC (%) | A | 32.0 | 32.1 | 32.1 | 32.1 | 32.1 |
| | T | 33.4 | 33.4 | 33.4 | 33.4 | 33.4 |
| | C | 17.7 | 17.7 | 17.7 | 17.7 | 17.7 |
| | G | 16.8 | 16.8 | 16.8 | 16.8 | 16.8 |
| | GC | 34.6 | 34.5 | 34.5 | 34.5 | 34.5 |
| SSC (%) | A | 34.9 | 34.9 | 34.9 | 34.9 | 34.9 |
| | T | 34.5 | 34.6 | 34.6 | 34.6 | 34.3 |
| | C | 16.1 | 16.1 | 16.1 | 16.1 | 16.1 |
| | G | 14.6 | 14.4 | 14.4 | 14.4 | 14.6 |
| | GC | 30.6 | 30.5 | 30.5 | 30.5 | 30.7 |
| IR (%) | A | 28.9 | 29.0 | 29.0 | 29.0 | 29.0 |
| | T | 29.1 | 29.0 | 29.0 | 29.0 | 29.1 |
| | C | 21.8 | 21.8 | 21.8 | 21.8 | 21.8 |
| | G | 21.8 | 20.1 | 20.1 | 20.1 | 20.2 |
| | GC | 41.9 | 42.0 | 42.0 | 42.0 | 42.0 |
| Overall length (%) | A | 31.3 | 31.3 | 31.3 | 31.3 | 31.3 |
| | T | 32.0 | 32.0 | 32.0 | 32.0 | 32.0 |
| | C | 18.7 | 18.7 | 18.7 | 18.7 | 18.7 |
| | G | 18.0 | 18.0 | 18.0 | 18.0 | 18.1 |
| | GC | 36.7 | 36.7 | 36.7 | 36.7 | 36.7 |

reported in other cp genomes (_Raubeson et al., 2007_; _Wu et al., 2017_). GUG start codons have been reported in tobacco, but they are very rare in eukaryotic genomes (_Kuroda et al., 2007_). When GUG was the start codon of a protein, it was still translated as Met because of the separate tRNA used for initiation. Furthermore, the codon usage patterns of the 58 distinct protein-coding genes in the five plastomes were examined, and the plastomes of _P. kangdingensis_, _P. pseudoglauca_, and _P. schneideri_ were consistent, with a length of 75,990 bp and encoding 25,330 codons, while those of _P. cathayana_ and _P. xiangchengensis_ were 75,864 and 75,840 bp in size and encoded 25,288 and 25,280 codons, respectively, as presented in Table S3. Coding ending with A and T/U had RSCU values >1 for the five _Populus_ cp genomes, indicating that they were used more frequently than synonymous codons and may play major roles in the A+T bias of entire cp genomes. There was a general excess of A- and U-ending codons. All three stop codons were present, with UAA being the most frequently used among the five plastomes (Table S3). In addition, leucine (Leu, 10.67%, 10.65%, 10.65%, 10.65%, and 10.65%) and cysteine (Cys, 1.14%) were the most and least commonly coded amino acids, respectively, among the five plastomes (Table S3; Fig. 3).

## Comparative analysis of the five _Populus_ plastomes

In this study, the cp genomes of the five _Populus_ species were well conserved, and no gene organization rearrangement occurred when _P. xiangchengensis_ was used as a reference (Figs. 4 and 5). LSC, SSC, and IR sections of the three _Populus_ species of _P. kangdingensis_,

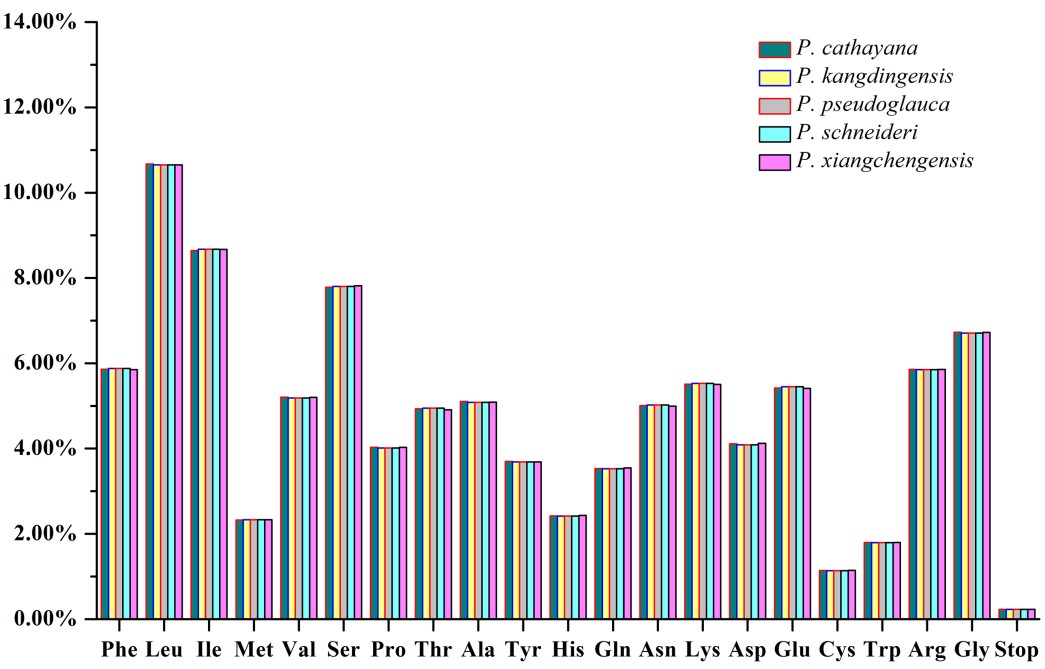

**Figure 3 Amino acid frequencies of the protein-coding sequences of the five plastomes.**

*P. pseudoglauca*, and *P. schneideri* were highly conserved and smaller than those of *P. cathayana*, while the IR regions were larger than those of *P. xiangchengensis*. Detailed comparisons of the IR-SSC and IR-LSC boundaries among the cp genomes of the five species are presented in Fig. 6. Two complete or fragmented copies of *rpl22* and *ycf1* were located at the boundaries between the LSC or SSC regions and IR regions among the five *Populus* plastomes. The *rpl22* gene crossed the IR-LSC with only one bp variation in sequence length among the five plastomes. Parts of the *ycf1* gene (15 (*P. cathayana*) -158 bp (*P. kangdingensis*, *P. pseudoglauca*, and *P. schneideri*)) were found in the SSC region at the IRb-SSC junction, whereas a portion of the *ycf1* gene (1,689 (*P. xiangchengensis*) to 1,707 bp (*P. cathayana*)) was present in both IRs. A 61 bp overlap between *ycf1* and *ndhF* was found in *P. kangdingensis*, *P. pseudoglauca*, and *P. schneideri*. The *Pi* values within the slide window of 600 bp in the five plastomes varied from 0.00001 to 0.00335 (Table 3), with a mean of 0.00210. However, nine highly variable loci (*Pi* > 0.01), including *trnG-atpA*, *psbZ-trnfM*, *trnL-ndhJ*, *ndhC-trnV*, *ycf4-cemA*, *trnN-trnR*, *ycf1*, *ccsA+ccsA-ndhD,* and *trnR-trnN*, were located in the five *Populus* plastomes (Fig. 7). Among these regions, *trnG-atpA*, *psbZ-trnfM*, *trnL-ndhJ*, *ndhC-trnV*, and *ycf4-cemA* were located in the LSC region, *ycf1* and *ccsA+ccsA-ndhD* were in the SSC region, and *trnN-trnR* and *trnR-trnN* were in the IR regions.

## Number and forms of mutations

We investigated SNPs, the most abundant type of mutation, in the five plastomes, with *P. xiangchengensis* as the reference. In gene-coding regions, we detected 70 SNPs in the comparative combination of *P. cathayana–P. xiangchengensis*, including 33 transition (Ts)

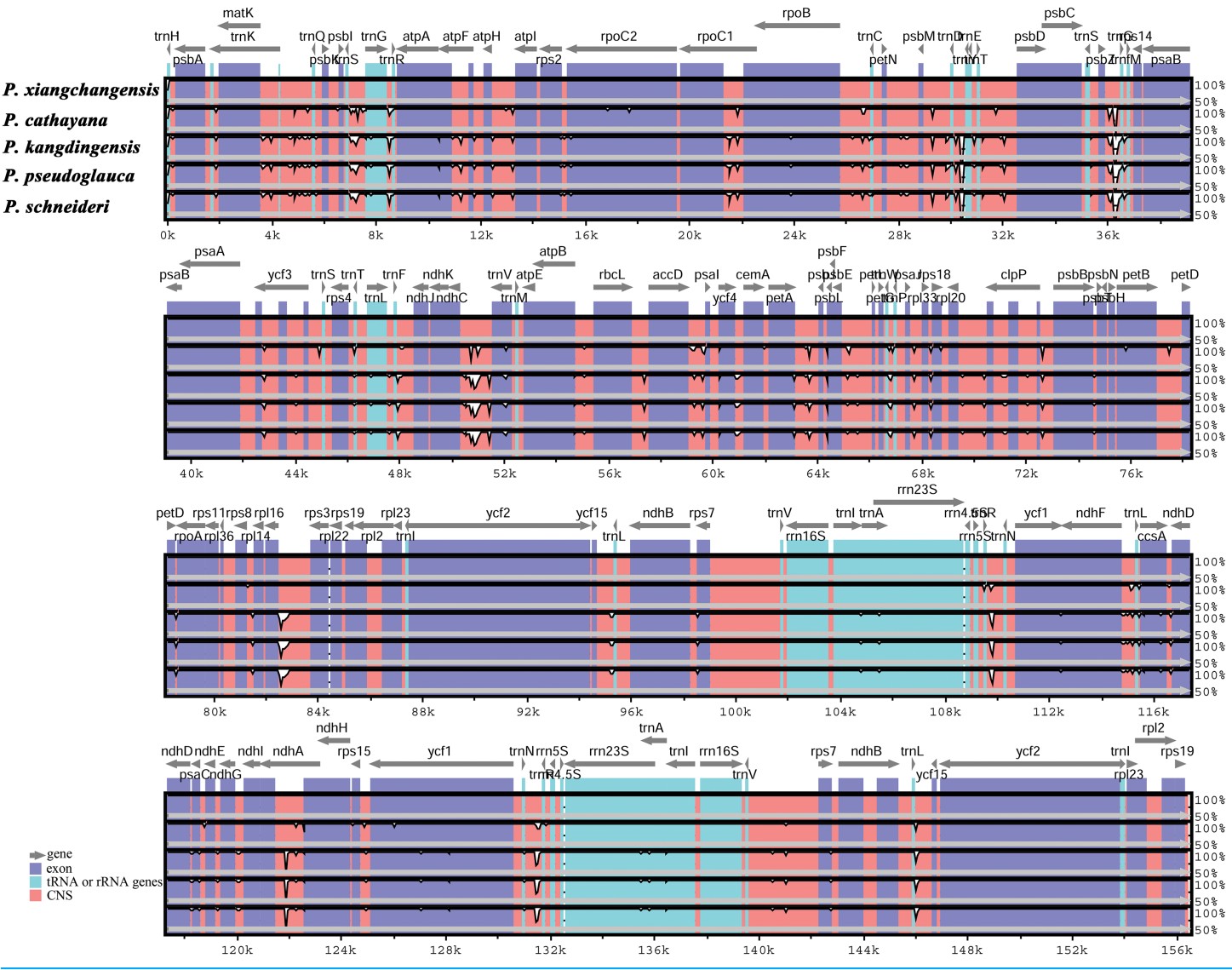

**Figure 4** Whole chloroplast genome alignments for five *Populus* species using the mVISTA program, with *P. xiangchengensis* as the reference. The *Y*-scale indicates identity from 50% to 100%, Gray arrows indicate the position and direction of each gene. Red indicates non-coding sequences (CNS), blue indicates the exons of protein-coding genes (exon) and line green indicates the tRNA or rRNA genes.

and 37 transversion (Tv) SNPs, as well as 160 (97 Ts and 63 Tv), 166 (101 Ts and 65 Tv) and 164 (99 Ts and 65 Tv) SNPs in the plastomes of *P. kangdingensis–P. xiangchengensis*, *P. pseudoglauca–P. xiangchengensis,* and *P. schneideri–P. xiangchengensis*, respectively (Table 4). Furthermore, 106 (38 Ts and 68 Tv), 323 (130 Ts and 193 Tv), 316 (130 Ts and 186 Tv), and 314 (131 Ts and 183 Tv) SNPs were detected in noncoding regions among the four comparative combinations, respectively (Table S4).

In our study, a total of six small inversions (*petA-psbJ*, *ndhC-trnV*, *trnN-trnR*, *ccsA-ndhD*, *ndhD-psaC,* and *ndhF-trnL*) were identified based on the sequence alignment of the five complete cp genomes (Fig. 8). The small inversions from *ndhC-trnV* and *ndhD-psaC* occurred in only *P. xiangchengensis*, those from *ndhF-trnL* occurred in
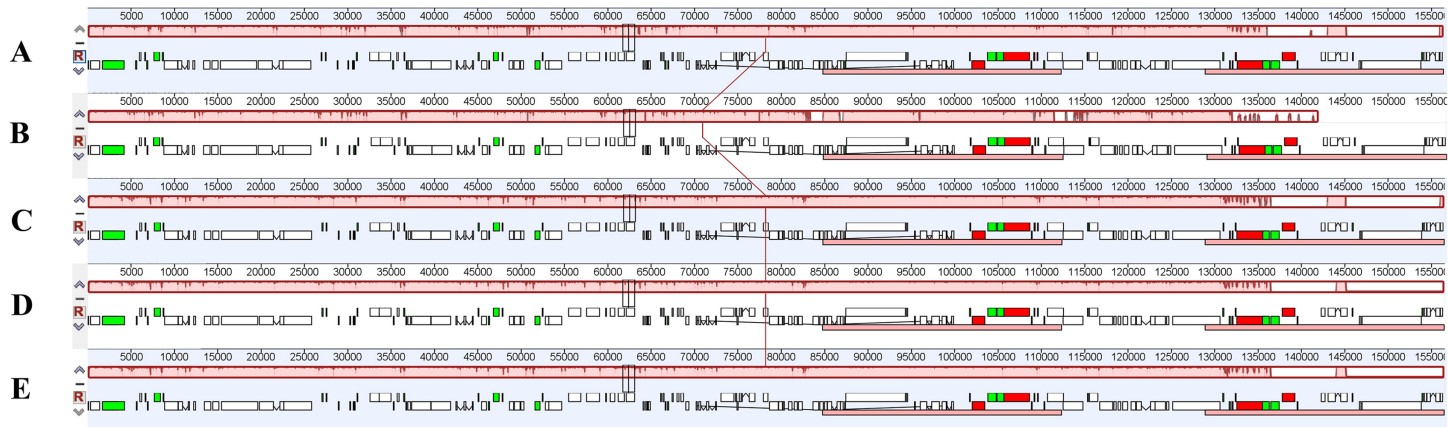

**Figure 5  Mauve alignment of the cp genomes of five *Populus* species.** (A) Cp genome of *P. xiangchengensis*; (B) Cp genome of *P. cathayana*; (C) Cp genome of *P. kangdingensis*; (D) Cp genome of *P. pseudoglauca*; (E) Cp genome of *P. schneideri*.

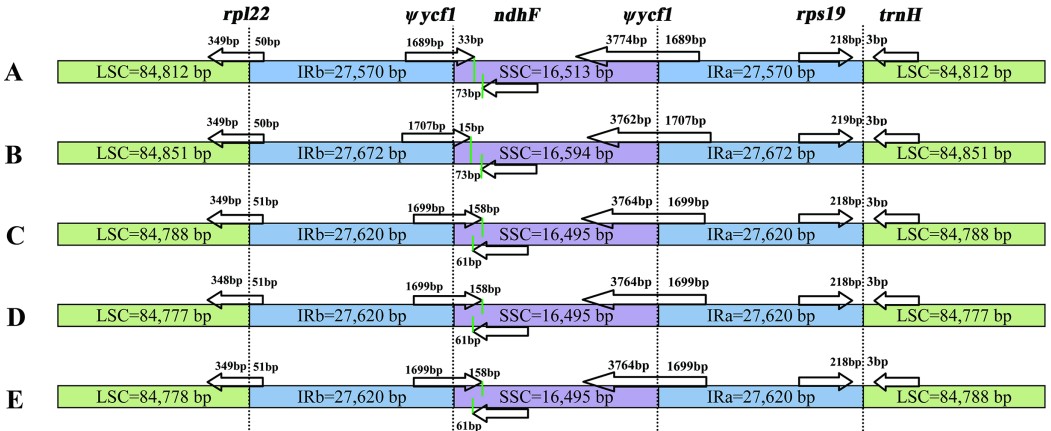

**Figure 6  Comparison of LSC, SSC, and IR region borders among cp genomes of five *Populus* species.** (A) Cp genome of *P. xiangchengensis*; (B) Cp genome of *P. cathayana*; (C) Cp genome of *P. kangdingensis*; (D) Cp genome of *P. pseudoglauca*; (E) Cp genome of *P. schneideri*.

**Table 3  Pairwise nucleotide divergences of the five *Populus* plastomes.**

| Species | *P. cathayana* | *P. kangdingensis* | *P. pseudoglauca* | *P. schneideri* | *P. xiangchengensis* |
|---|---|---|---|---|---|
| *P. cathayana* | – | | | | |
| *P. kangdingensis* | 0.00124 | – | | | |
| *P. pseudoglauca* | 0.00333 | 0.00326 | – | | |
| *P. schneideri* | 0.00335 | 0.00327 | 0.00003 | – | |
| *P. xiangchengensis* | 0.00333 | 0.00325 | 0.00001 | 0.00002 | – |

*P. pseudoglauca* and *P. schneideri*, those from *trnN-trnR* occurred in *P. kangdingensis*, *P. pseudoglauca* and *P. schneideri*, and those from *ccsA-ndhD* occurred in the four species other than *P. cathayana*, while the inversion from *petA-psbJ* occurred in the four species other than *P. xiangchengensis* (Fig. 8).

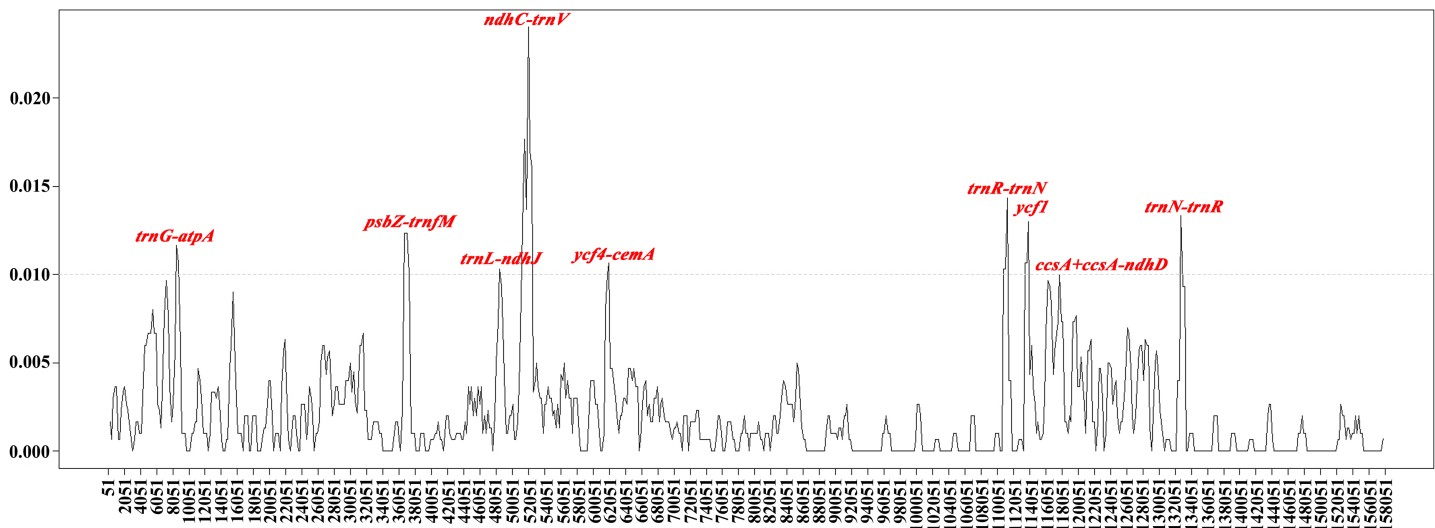

**Figure 7 Sliding window analyses of the whole plastomes of five *Populus* species.** The genetic divergence among the plastomes of *P. cathayana*, *P. kangdingensis*, *P. pseudoglauca*, *P. schneideri* and *P. xiangchengensis* was calculated with DnaSP 5.0 software (window length: 600 bp, step size 200 bp). *X*-axis: position of the midpoint of a window, *Y*-axis: nucleotide diversity of each window.

**Table 4 Transitions (Ts) and transversions (Tv) in the protein-coding regions of the four compare combinations with the plastome of *P. xiangchengensis*.**

| | Ts | | Tv | | | | Total |
|---|---|---|---|---|---|---|---|
| | A–G | C–T | A–T | A–C | T–G | G–C | |
| *P. cathayana–P. xiangchengensis* | 14 | 19 | 13 | 7 | 9 | 8 | 70 |
| *P. kangdingensis–P. xiangchengensis* | 42 | 55 | 12 | 14 | 24 | 13 | 160 |
| *P. pseudoglauca–P. xiangchengensis* | 44 | 57 | 12 | 15 | 24 | 14 | 166 |
| *P. schneideri–P. xiangchengensis* | 43 | 56 | 12 | 15 | 24 | 14 | 164 |

## Synonymous (*K*s) and nonsynonymous (*K*a) substitution rate analysis

In this study, ratios of nonsynonymous (*K*a) vs synonymous (*K*s) substitutions were calculated for 78 shared unique protein coding genes in *P. cathayana*, *P. kangdingensis*, *P. pseudoglauca,* and *P. schneideri*, with *P. xiangchengensis* as the reference. Among these genes, only 19 protein-coding genes had *K*a/*K*s values (Fig. 9; Table S5). The *K*a/*K*s values of the remaining protein-coding genes could not be calculated because *K*a or *K*s was equal to 0, indicating that these sequences were conserved without nonsynonymous or synonymous nucleotide substitution. The *K*a/*K*s ratios of all genes except *rpoC2* in *P. pseudoglauca* (1.00903) and the *rbcL* gene in *P. kangdingensis* (2.26407), *P. pseudoglauca* (2.26407), and *P. schneideri* (2.26407) were less than 1 (Fig. 9).

## SSR and long repeat analysis

With MISA, a total of 170 SSR loci were detected, of which mononucleotide repeats (P1) comprised 148 (87.06%) of all SSRs and all of the mononucleotides composed of poly A (polycytosine) and poly T (polythymine) repeats (Table 5). Within the five plastomes,

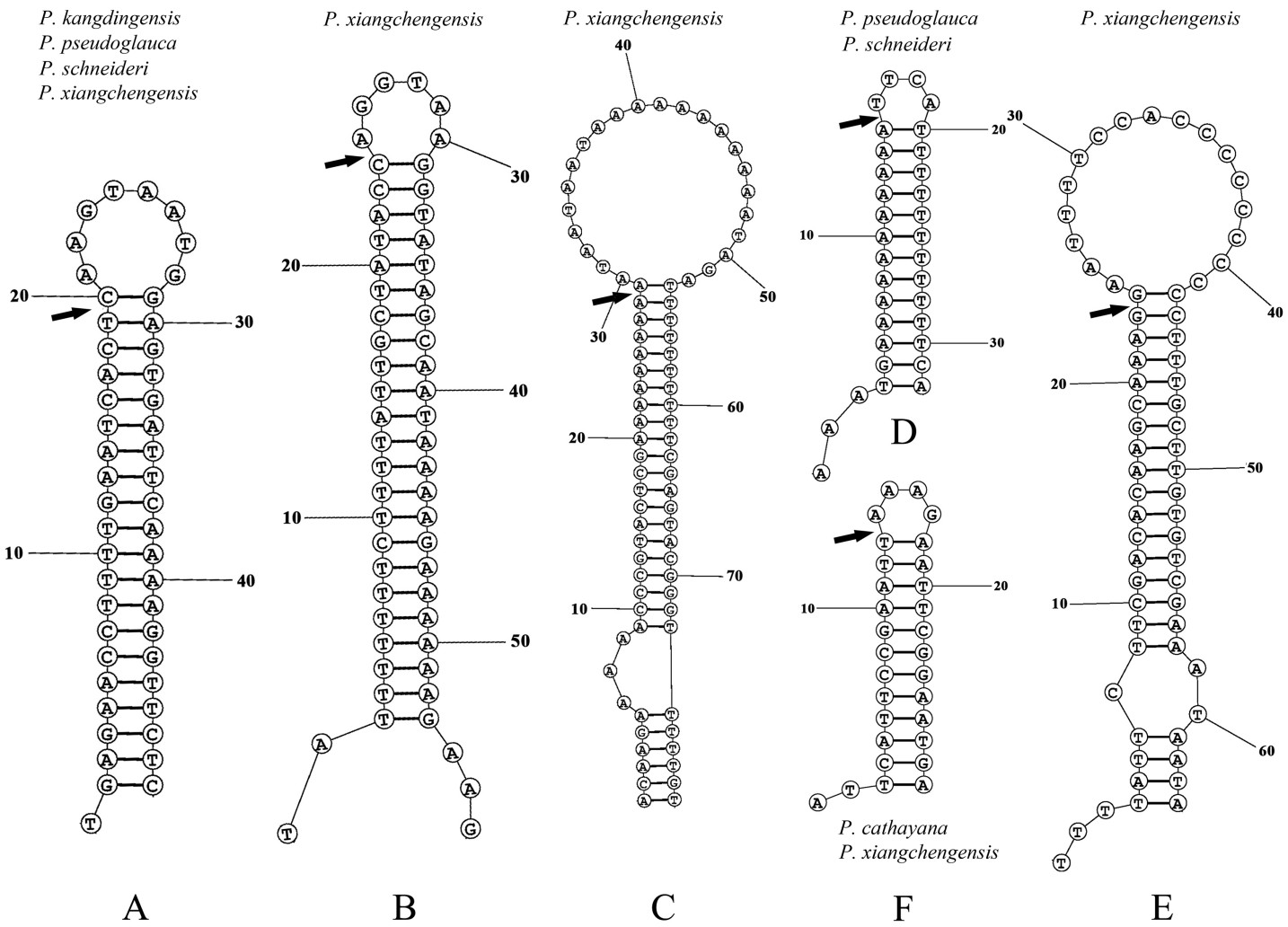

**Figure 8 Predicted hairpin loops of inversions in the five plastomes of *Populus*.** The structures of hairpin loops in the of the (A) *ccsA-ndhD*, (B) *ndhC-trnV*, (C) *ndhD-psaC*, (D) *ndhF-trnL*, (E) *petA-psbJ*, and (F) *trnN-trnR* regions were drawn with RNAstructure. The arrows in the figure indicate the break points in inversion events.

SSR loci were primarily located in the LSC region, followed by the SSC region. A total of 15 SSR loci were detected in the protein-coding genes *rpoB*, *rpoC2*, and *rps8*, with all others situated in intergenic spacers and introns (Table S6). A total of 28, 39, 39, 39, and 25 SSR loci were detected in the *P. cathayana*, *P. kangdingensis*, *P. pseudoglauca*, *P. schneideri*, and *P. xiangchengensis* cp genomes, respectively (Table 5). The corresponding numbers of these repeats in *P. kangdingensis*, *P. pseudoglauca*, and *P. schneideri* matched each other and consisted of 33 P1, two dinucleotide (P2) and four compound (C) repeats. Comparison among the five plastomes revealed that four P1 loci were found in only *P. xiangchengensis*, one C locus was found only in *P. cathayana*, and 15 SSR loci (11 P1, 1 P2, and 3C repeats) were detected in the plastomes of *P. kangdingensis*, *P. pseudoglauca*, and *P. schneideri* (Table S6).

In the plastomes of the five *Populus* species, we found 58 repeats in *P. cathayana*, which was a higher number than those found in the other four species (49, 48, 48, and 48 repeats,

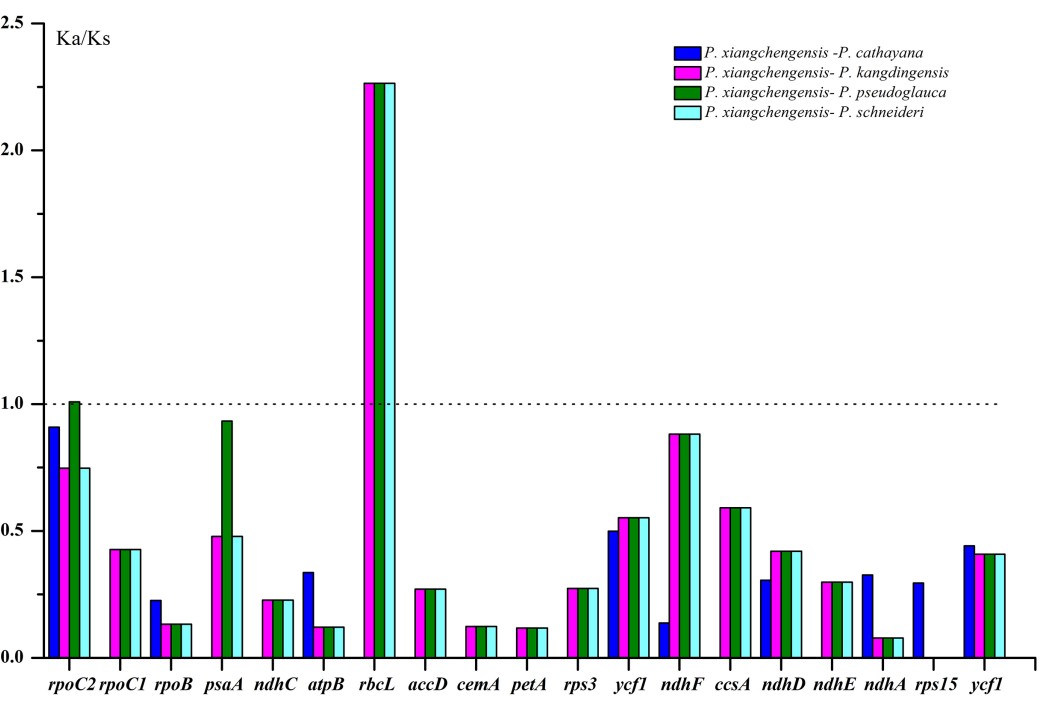

**Figure 9 *Ka/Ks* values of 19 protein-coding genes of the four comparative combinations.** *K*a, non-synonymous; *K*s, synonymous.

**Table 5 Statistics of chloroplast SSRs detected in five *Populus* plastomes.**

| SSR type | | *P. cathayana* | *P. kangdingensis* | *P. pseudoglauca* | *P. schneideri* | *P. xiangchengensis* |
|---|---|---|---|---|---|---|
| P1 | (A)12 | 3 | 9 | 9 | 9 | 3 |
| | (A)13 | 5 | 5 | 5 | 5 | 4 |
| | (A)14 | 2 | 2 | 2 | 2 | 3 |
| | (A)15 | 2 | 0 | 0 | 0 | 1 |
| | (A)16 | 1 | 2 | 2 | 2 | 0 |
| | (A)17 | 0 | 1 | 1 | 1 | 0 |
| | (T)12 | 4 | 5 | 5 | 5 | 3 |
| | (T)13 | 3 | 3 | 3 | 3 | 3 |
| | (T)14 | 3 | 1 | 1 | 1 | 1 |
| | (T)15 | 1 | 1 | 1 | 1 | 3 |
| | (T)16 | 2 | 4 | 4 | 4 | 1 |
| | (T)17 | 0 | 0 | 0 | 0 | 1 |
| | ALL | 26 | 33 | 33 | 33 | 23 |
| P2 | TA/AT | 1 | 2 | 2 | 2 | 1 |
| C | | 1 | 4 | 4 | 4 | 1 |
| Total | | 28 | 39 | 39 | 39 | 25 |

**Note:**
P1, mononucleotide repeats; P2, dinucleotide repeats; C, compound repeats.
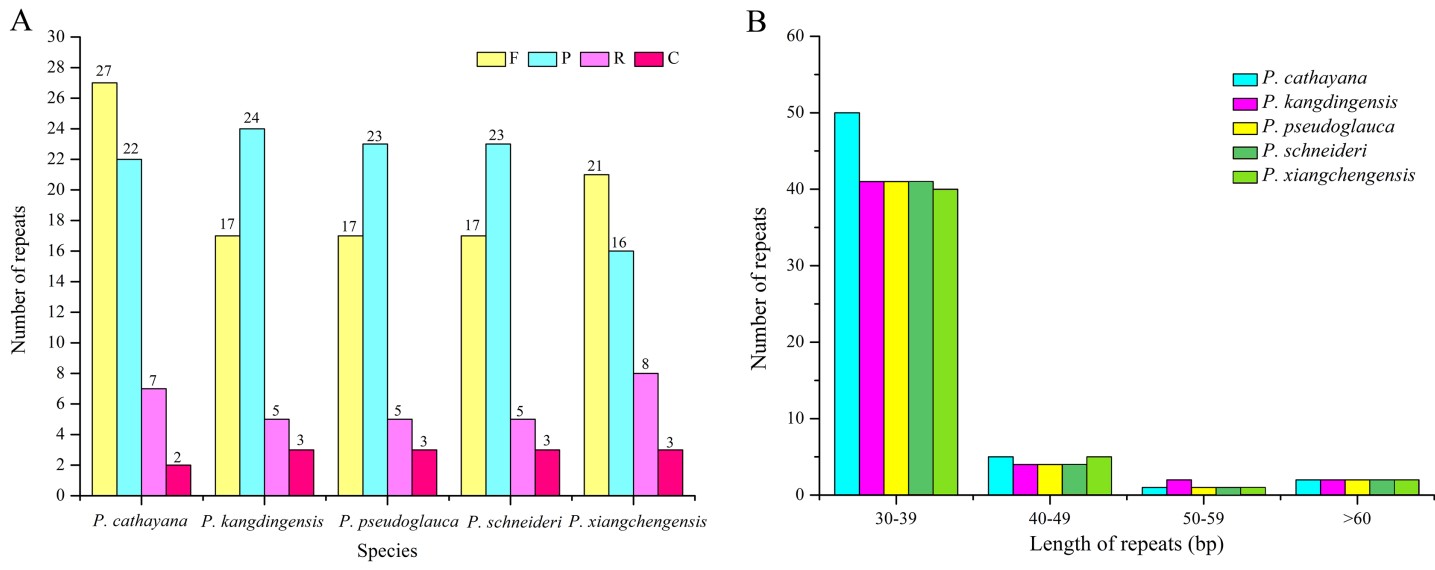

**Figure 10 Comparison of long repeats among five *Populus* plastomes.** (A) Number of each repeat type; F, forward repeats; P, palindromic repeats; R, reverse repeats and C, complement repeats (B) frequency of each repeat type by length.

respectively) (Fig. 10A). *P. pseudoglauca* and *P. schneideri* shared the same number and types of repeats (17 forward repeats (F), 23 palindromic repeats (P), five reverse repeats (R), and three complement repeats (C)) (Fig. 10A). The majority of repeats (84.86%) varied from 30 to 39 bp in length (Fig. 10B). Variation in the number of repeat sequences situated in the four parts of the plastome was observed between species (Table S7).

## Phylogenetic analysis based on the cp genome

Seven data partitions (complete cp genomes, LSC, SSC, IR, LSC+SSC, LSC+SSC+IR region, and protein coding regions) from 22 Salicaceae cp genomes were used to construct phylogenetic trees (Figs. S1–S6; Fig. 11). However, the best resolution in phylogenetic relationships was achieved using full cp genome sequences; thus, we discuss phylogenetic relationships mainly based on Fig. 11. All of *Populus* was divided into four main highly supported clades (Fig. 11). Three species of the section *Turanga* were clade I members. Clade II included seven species (*P. adenopoda, P. alba, P. davidiana, P. qiongdaoensis, P. rotundifolia, P. tremula,* and *P. tremula × alba*) in section *Populus* and one species in section *Aigeiros* (*P. nigra*). Clade III consisted of three species in section *Tacamahaca* (*P. kangdingensis, P. schneideri,* and *P. yunnanensis*) and two species in section *Leucoides* (*P. lasiocarpa* and *P. pseudoglauca*). Clade IV included the four species in section *Tacamahaca* (*P. balsamifera, P. cathayana, P. trichocarpa,* and *P. xiangchengensis*), one species in section *Aigeiros* (*P. fremontii*) and one species in section *Leucoides* (*P. wilsonii*). Our results showed that *P. kangdingensis, P. pseudoglauca,* and *P. schneideri* were in clade III, while *P. xiangchengensis* formed a sister relationship with 100% bootstrap support to *P. cathayana* in clade IV.

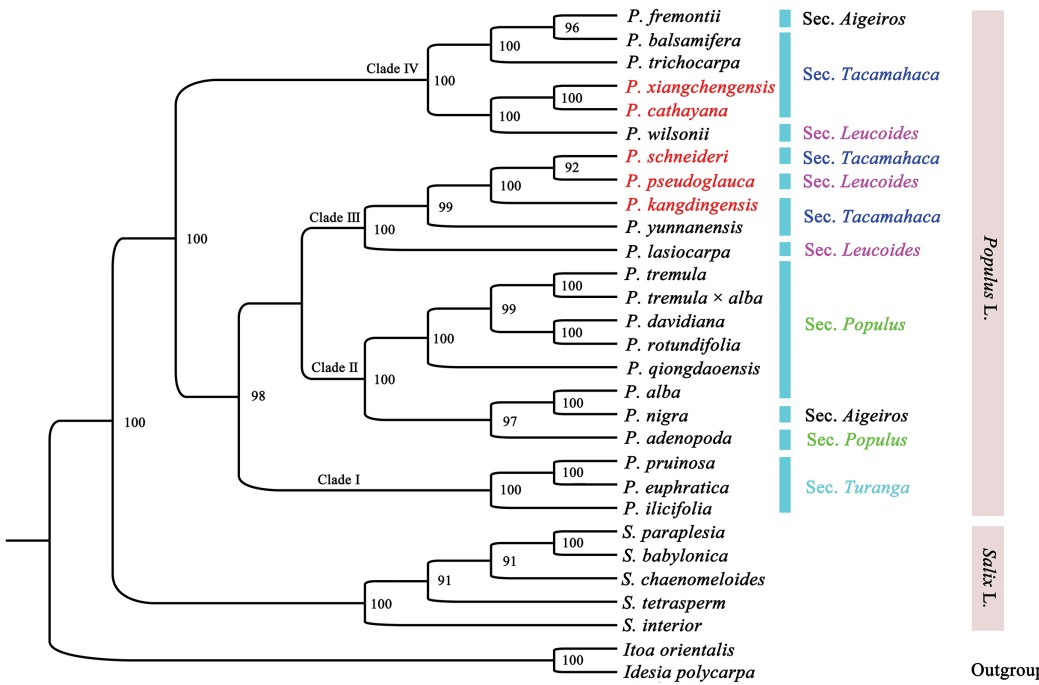

**Figure 11 Molecular phylogenetic tree of 27 species in the family *Salicaceae* inferred from ML analyses based on the complete plastome sequence.** The numbers associated with each node are bootstrap support values.           

## DISCUSSION

In the present study, we compared five *Populus* plastomes, all of which assembled into single, circular, double-stranded DNA sequences presenting a typical quadripartite structure with a length of 156,465 to 156,789 bp, similar to most *Populus* cp genomes (*Wang et al., 2016*; *Zhang & Gao, 2016*; *Zheng et al., 2016*; *Han, Wang & Liu, 2017*). The GC contents in the LSC, SSC and IR regions were consistent among the three species *P. kangdingensis*, *P. pseudoglauca,* and *P. schneideri*, and the high GC content in the IR regions was possibly due to the presence of four ribosomal RNA sequences in these regions (*Yang et al., 2016*). In addition, LSC, SSC, and IR sections of the three *Populus* species were highly conserved and smaller than those of *P. cathayana*, while the IR regions were larger than those of *P. xiangchengensis*. IRs are the most conserved regions of the cp genome, and the construction and expansion of IR borders are common evolutionary events and the major reasons for size differences between cp genomes (*Shen et al., 2017*; *Wu et al., 2018*).

The whole aligned sequences revealed surprisingly low divergence; however, nine regions (*trnG-atpA, psbZ-trnfM, trnL-ndhJ, ndhC-trnV, ycf4-cemA, trnN-trnR, ycf1, ccsA+ccsA-ndhD,* and *trnR-trnN*) displayed high variation. Further work is still necessary to determine whether these nine variable loci could be used in phylogenetic analyses of related *Populus* species or as potential molecular markers for population genetics and phylogenetics.

Understanding nucleotide substitution rates is of fundamental importance in molecular evolution (*Muse & Gaut, 1994*). During the process of searching for SNPs,
we found that the cp genome sequences of *P. kangdingensis*, *P. schneideri,* and *P. pseudoglauca* had similar numbers of mutations, while *P. cathayana* had a smaller number of mutations relative to *P. xiangchengensis.* Therefore, the phylogenetic relationships of these five species may be affected by different mutation modes (*Yang et al., 2016*). Furthermore, we found that the numbers and types of SSRs and long repeats of the three species *P. kangdingensis*, *P. schneideri,* and *P. pseudoglauca* were basically identical. These SSR repeats and long repeats could be used to examine genetic structure, diversity, differentiation and relative species in future studies.

The *K*a and *K*s nucleotide substitution patterns are very important markers in gene evolution studies (*Kimura, 1979*). The *K*a/*K*s ratio is indicative of changes in selective pressures. *K*a/*K*s values >1, =1, and <1 indicate positive selection, natural evolution and purifying selection affecting the coding portions, respectively (*Sharp & Li, 1987*; *Yang & Bielawski, 2000*; *Lawrie et al., 2013*). However, the *K*a/*K*s ratio is less than one in most protein-coding regions (*Makalowski & Boguski, 1998*). When we compared the 78 common unique protein coding genes, the *K*a/*K*s ratios were less than one, except the ratio for the gene *rpoC2* in one pairwise comparison of *P. pseudoglauca–P. xiangchengensis* and the ratio for the gene *rbcL* in the pairwise comparisons of *P. kangdingensis–P. xiangchengensis*, *P. pseudoglauca–P. xiangchengensis,* and *P. schneideri–P. xiangchengensis*, indicating that these two genes are undergoing positive selection and that at least some of the mutations concerned must be advantageous.

Small inversions in the cp genome of angiosperms are ubiquitous and commonly associated with a hairpin secondary structure (*Kim & Lee, 2005*; *Catalano, Saidman & Vilardi, 2009*). A distinctive feature of these inversions is that they are flanked by IRs that range from 8 to 50 bp, such that the IRs form the stem and the segment between them forms the loop (*Catalano, Saidman & Vilardi, 2009*). These small inversions are generally recognized by pairwise comparisons between sequences. In our study, six small inversions were discovered based on the sequence alignment of the five complete cp genomes. Two small inversions from *ndhC-trnV* and *ndhD-psaC* were only present in *P. xiangchengensis*, while three from *ccsA-ndhD*, *ndhF-trnL* and *trnN-trnR* were shared by *P. pseudoglauca* and *P. schneideri*. These small inversion regions will provide abundant information for marker development in phylogenetic analyses of related *Populus* species. Small inversions in the *ccsA-ndhD* and *petA-psbJ* intergenic regions have been reported in other studies (*Song et al., 2015*, *2016*; *Dong et al., 2017*). However, small inversions of noncoding sequences may influence sequence alignment and character interpretation in phylogeny reconstructions, so caution is necessary when using cp noncoding sequences for phylogenetic analysis.

The cp genome is widely employed to study evolution through phylogenetics, and it has been suggested to be useful for phylogenetic reconstruction at low taxonomic levels (*Zhang, Ma & Li, 2011*; *Ma et al., 2014*; *Yang, Li & Li, 2014*; *Zhang et al., 2016*). It has also been postulated to be a potential ultrabarcode or organelle-scale barcode for taxonomically complex groups (*Kane et al., 2012*). The key interest of the current study is to resolve previous phylogenetic controversies in *Populus* (*Zhao, 1994*; *Liu & Fu, 2004*; *Chen et al., 2007*; *Wan et al., 2009*; *Wang, 2012*) by using complete cp genome

sequences. The results revealed that *P. nigra* (section *Aigeiros*) was nested among members of the section *Populus*, which is supported by previous studies (*Rajora & Dancik, 1995*; *Hamzeh & Dayanandan, 2004*). Both previous studies found that *P. nigra* showed higher similarities to *P. alba* than to other species.

The position of *P. pseudoglauca* confirms the previously published phylogeny described by *Chao & Liu (1991)*, in which *P. pseudoglauca* was classified into section *Tacamahaca* according to fossil evidence, paleogeography, paleoclimate, and modern distribution. The species *P. schneideri*, which is distributed in the western Sichuan Plateau at altitudes of 3,000–4,000 m, has remained a topic of debate among scientists. According to its morphology, it is similar to *P. cathayana* (*Fang, Zhao & Skvortsov, 1999*). *Wan et al. (2013)* suggested that *P. schneideri* is generally closer to *P. cathayana* than *P. kangdingensis*, and it is a natural hybrid between the ancestors of *P. cathayana* and *P. kangdingensis* based on cpDNA and nuclear DNA sequence data as well as amplified restriction fragment polymorphism analyses. Other studies considered *P. schneideri* to be a variety of *P. kangdingensis* based on morphological traits (*Chao & Liu, 1991*; *Yu et al., 2003*; *Liu & Fu, 2004*). *Chen et al. (2007)* suggest that *P. schneideri* is generally more highly related to *P. kangdingensis* than to *P. cathayana* based on cpSSR analysis. Our data revealed that *P. schneideri* had a close relationship with *P. pseudoglauca* and *P. kangdingensis*. *P. schneideri* and *P. kangdingensis* are both unique to the western Sichuan Plateau, and they share similar altitude and habitat requirements (*Yu et al., 2003*). In addition, the plastome characteristics, SNP variations, SSR and long repeats and small inversions all showed that *P. kangdingensis*, *P. pseudoglauca,* and *P. schneideri* had similar structure and variation models. Therefore, we speculated that these three species may be the same species; however, more population data are needed for further analyses. *P. xiangchengensis* was a sister to *P. cathayana*, as revealed by cp genome sequence analysis, which did not support the viewpoint that *P. xiangchengensis* is a natural hybrid species of either *P. schneideri* and *P. pseudoglauca* or *P. kangdingensis* and *P. pseudoglauca*. It is our hope that the five plastomes will provide useful resources for better understanding the phylogeny and relationships of the genus *Populus*.

## CONCLUSIONS

This study reports the comparative analysis of five *Populus* cp genome sequences with detailed gene annotation. Comparing the five plastomes showed that the plastomes were similar in structure and had a high degree of synteny. Nine divergent regions (*trnG-atpA, psbZ-trnfM, trnL-ndhJ, ndhC-trnV, ycf4-cemA, trnN-trnR, ycf1, ccsA+ccsA-ndhD,* and *trnR-trnN*) were identified and may be utilized as potential molecular markers for population genetic and phylogenetic studies in *Populus*. Furthermore, among the five cp genomes, *P. kangdingensis*, *P. pseudoglauca,* and *P. schneideri* showed little difference in their SNP loci and SSRs. The results of the phylogenetic analyses showed that *P. schneideri* had the closest affinity to *P. kangdingensis* and was a sister to *P. pseudoglauca*, while *P. cathayana* had a close relationship with *P. xiangchengensis*. The characterization of these five plastomes will provide useful resources for better understanding the phylogeny and relationships of the genus *Populus*.

## ACKNOWLEDGEMENTS

We would like to thank Prof. Wuyuan Yin for his help with sample collection and species identification. We are grateful to Dr. Hui Zhan for language review. We also thank the editor and the two anonymous reviewers for their constructive comments, which helped us improve this manuscript.

### Funding

This work was supported by the National Natural Science Foundation of China (Grant No. 31460205, 31860219, 31360184) and the Forestry Public Benefit Research Program (No. 201104076). The funders had no role in study design, data collection and analysis, decision to publish, or preparation of the manuscript.

### Grant Disclosures

The following grant information was disclosed by the authors:
National Natural Science Foundation of China: 31460205, 31860219, 31360184.
Forestry Public Benefit Research Program: 201104076.

### Competing Interests

The authors declare that they have no competing interests.

### Author Contributions

- Dan Zong conceived and designed the experiments, performed the experiments, analyzed the data, prepared figures and/or tables, authored or reviewed drafts of the paper, approved the final draft.
- Anpei Zhou performed the experiments, prepared figures and/or tables, authored or reviewed drafts of the paper, approved the final draft.
- Yao Zhang performed the experiments, approved the final draft.
- Xinlian Zou performed the experiments, approved the final draft.
- Dan Li performed the experiments, approved the final draft.
- Anan Duan conceived and designed the experiments, contributed reagents/materials/analysis tools, authored or reviewed drafts of the paper, approved the final draft.
- Chengzhong He conceived and designed the experiments, contributed reagents/materials/analysis tools, authored or reviewed drafts of the paper, approved the final draft.

### DNA Deposition

The following information was supplied regarding the deposition of DNA sequences:
The complete chloroplast genome sequences for the four Populus species are available at GenBank: MK267318, MK267310, MK267308, MK267305.

### Data Availability

The raw data are available in the Supplementary File.

## Supplemental Information

Supplemental information for this article can be found online at http://dx.doi.org/10.7717/peerj.6386#supplemental-information.

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
