# Peer review of "Characterization of the complete chloroplast genomes of five Populus species from the western Sichuan plateau, southwest China: comparative and phylogenetic analyses"

_PeerJ, doi:10.7717/peerj.6386_

## Round 0.1 · original submission · Minor Revisions

The reviewers are in agreement on this manuscript and have both indicated minor changes that will improve the paper. Please follow their recommendations.

·

Basic reporting

English in the reviewed manuscript is good, however there are some parts of the text which needs revision/clarification (e.g. L. 90, L. 247, L. 311-313, L.319-320, L.321-322)

I have noticed two mistakes in reference citation in L. 345 – there is no such position in reference list like Zheng 2016 and Wang 2016, there should be Zheng et al. 2016 and Weng et al. 2016, respectively. I recommend more exhaustive reference check.

The article is well structured. Tables and figures should be self-explanatory therefore I recommend to explain the meaning of abbreviations used in the caption of Figure 8 and within Table 5 and Table S3. Moreover, the caption of TableS4 should be improved - Synonymous (Ks) and nonsynonymous (Ka) substitution rate analysis of the five species, with P. xiangchengensis as the reference.

None of the reported cp genomes is accessible (yet) in GenBank (NCBI).

Experimental design

The reviewed paper reports the results of chloroplast genome sequencing of Populus species (P. cathayana, P. kangdingensis, P. pseudoglauca, P. schneideri and P. xiangchengensis). Complete chloroplast genome sequences were characterized and used for comparative and evolutionary studies. Based on the obtained molecular data authors propose the candidate chloroplast sequences suitable for further population genetic and phylogeographic studies. The presented chloroplast genomes help to resolve the phylogenetic relationships among Populus species

Objectives of the paper are clear and unambiguous. The methodology is correct and suitable for realization of the paper objectives. The results description is adequate and conclusions are generally well supported. Some additional minor flaws are marked within the manuscript or are addressed below.

Why did authors decide to perform the phylogenetic analyses based on the complete cp genomes instead of coding sequences only, although according to their statement (L.376-377): “…caution is necessary when using cp noncoding sequences for phylogenetic analysis.”?

How did the authors choose the model used for phylogenetic analyses in RAxML? Have you tested also other models for your data?

Validity of the findings

The reviewed paper is an example of an interesting and valuable study with well-planned and extensive but complementary methodological side, as well as clearly delineated objectives of the work. The described results perfectly correspond and are complementary to previous studies on history, evolution, and diversity of Populus. The applied methodology, high-throughput sequencing, provide a high-quality data with a number of applications. Here, authors not only report complete chloroplast genomes for five Populus species but also based on the molecular data revised the systematics of the genus. The observations included in the reviewed manuscript may become a valuable element of the discussion not only in case of studies on evolution and diversity of Populus but also for other closely related taxons.

Additional comments

The reviewed manuscript represents an interesting and valuable study which should be considered for publication in PeerJ but after minor revisions

Reviewer 2 ·

Basic reporting

The authors report the findings of a study to characterize and compare five Populus plastomes, of which four are reported here first. The article covers the relevant background to evaluate the findings. I found the article interesting and timely. I suggest a restructuring of the paper so that the reason for the study, a phylogenetic assessment of the study species, could be emphasised. It is only stated clearly in the second to last paragraph of the discussion in lines 382-384. I believe reporting the findings from phylogenetic relationships to structure/characterization of the plastomes would provide a clearer narrative for the paper. This would also increase the relevance of the findings. Moreover, the characteristic of the genomes are more interesting in light of the relationship of the species.
The paper generally needs some restructuring as parts of the results should be in methods or discussion (see detailed comments below.). The figures are well designed, but see comments below for a few possible improvements.

Experimental design

The study fits well in PeerJ’s scope and the experimental design is suitable to answer the main question of the study. I could not find any evidence that the assembled plastomes are/will be submitted to GenBank. You could include the GenBank numbers (or a placeholder if they have not been issued with accession numbers) in lines 185-186 or Table 1.

Validity of the findings

See comments above

Additional comments

Line 74 What do you mean with ‘100 and more Populus species’?
Line 90 replace hybridization with hybrid
Line 91-92 It would be useful to include what data the Wan et al. 2009 study used to suggest the hybrid parentage of P. xiangchengensis. It would also be easier to follow this if you could include a diagram or a table of the suggested but disagreeing hybridization events that lead to P. xiangchengensis.
Line 95 delete ‘the’ before section Tacamahaca
Line 99 ‘as the uniparental inheritance’ is unclear.
Line 109 I would delete ‘the’ and ‘database’ around GenBank
Line 110 You should make it clearer that four of the listed species are sequenced for this study while the fifth is from a study currently in press
Line 115 I find selective breeding and crossbreeding overlapping to use both. Maybe you could use ‘resources for breeding’?
Line 127 genomic instead of genome
Line 128 What do you mean properly storing the DNA? -20 or -80?
Line 134 Are the 15.0 Gb data total for the four species or 15 Gb each?
Line 185 Whi,le we accept that plastome is made of double-stranded DNA as far as I know you can’t really see this from next-gen data. I think it would be best to just delete it.
Line 194-195 should be discussion
Line 205-205 These characters of the cp genomes are usual so I don’t think this needs reporting.
Line 210-211 How did you code the ATC start codon?
Line 217-218 should be methods
Line 223 Why is frequency of leucine and cysteine interesting?
Line 228-229 should be discussion
Line 233-234 should be discussion
Line 241-245 This is unclear. The copy of ycf1 in IRb is too short to refer to as gene. I think it would be more suitable to use pseudogene or a truncated copy of ycf1. Parts of the ycf1 gene (or pseudogene) are found in the SSC 15-158 bp at the IRb – SSC junction. The portion of the gene 1689-1707 bp is present in both IRs.
Line 246-247 should be methods
Line 248 the mean of 0.00210 is not in Table 3
Line 248-250 should be discussion
Line 251 I don’t think there is a valid intergenic spacer between ycf1 and ndhF. Its presence is entirely dependent on the structure of the IR junctions and based on Figure 5 missing from 3 out of the 5 species.
Line 252-254 should be discussion
Line 257-264 I found this bit too hard to read. It would be easier to report a general description or ranges of the SNPs and reference the table with the details.
Line 265-266 should be discussion or methods
Line 268 identified instead of uncovered?
Line 276-282 should be discussion or methods
Line 282-285 I don’t know what the 85 comparison refers to
Line 288-290 should be discussion
Line 289 undergoing instead of suffered
Line 293-298 should be introduction or discussion
Line 300 I would delete ‘occurred with high frequency’
Line 306-310 these are already in a table
Line 314-316 should be discussion
Line 316-320 this is a bit unclear
Line 322-323 should be in discussion
Line 327 I don’t think you need ‘further’
Line 326-328 should be methods
Line 328 Do you mean downloaded from GenBank instead of public?
Line 360-362: I think this might be the other way round: the different mutation modes are affected by the phylogenetic relationship, as in closely related species show similar mutation modes.
Line 365 I think calling the repetitive elements of inversions IRs are misleading as IR normally refers to the inverted repeat region of the whole cp geome
Line 367-373 This is a word by word repeat of Line 266-273
Line 384-395 This is a word by word repeat of Line 329-339
Line 398-400 What is debated about P. schneideri. I assume it’s whether it is a hybrid or not, but it would be useful to state this again here.
Figure 6 I would make the labels larger
Figure 7 I would include the species in which each loops are found in the figure or in the caption.
Figure 9 The caption needs to include the meaning of F, P, R, C. Also the similar patterns for panel A and B are confusing. If possible use colours here.
Figure 10 I would label the sections Populus on the figure. The caption needs to explain the meaning of the support values and the method the tree was generated.
Table 3 Is the pairwise nucleotide divergence averaged from the 600-bp rolling window over the length of the pairwise alignments?

---

## Round 0.2 · accepted · Accept

Thank you for incorporating the suggested amendments. The manuscript is now sufficiently revised to be ready for publication.

#